# Database of virtual objects to be used in psychological research

**Deian Popic, Simona G. Pacozzi, Corinna S. Martarelli** ID *

Faculty of Psychology, Swiss Distance University Institute, Brig, Switzerland

* corinna.martarelli@fernuni.ch

**Data Availability Statement:** All relevant data are available on the Open Science Framework (osf.io/q658a/).

**Funding:** The authors received no specific funding for this work.

## Abstract

Although many visual stimulus databases exist, to our knowledge, none includes 3D virtual objects, that can directly be used in virtual reality (VR). We present 121 objects that have been developed for scientific purposes. The objects were built in Maya, and their textures were created in Substance Painter. Then, the objects were exported to an FBX and OBJ format and rendered online using the Unreal Engine 4 application. Our goal was to develop the first set of high-quality virtual objects with standardized names, familiarity, and visual complexity. The objects were normed based on the input of 83 participants. This set of stimuli was created for use in VR settings and will facilitate research using VR methodology, which is increasingly employed in psychological research.

## Introduction

Since virtual reality (VR) technology emerged as research method in the 1980s, it has evolved in different ways to become less expensive, more user-friendly, and more immersive. Current fully immersive VR environments presented via a head-mounted display are becoming increasingly realistic. They conform to human vision (e.g., motion parallax), and they allow for motion and physical interaction with the virtual world in a similar manner as with reality. Immersive VR provides the unique feeling of being physically present in a computer-generated virtual environment, as the user is fully immersed in the virtual world.

During the last two decades, VR has been increasingly used in psychological research. In a seminal paper, Blascovich et al. [1] discussed the trade-off between experimental control and external validity that characterizes research methods in psychology. They suggested that immersive VR might one day eliminate the trade-off entirely. For instance, in the field of visual memory research, encoding, storage, and retrieval processes are investigated in conventional laboratory settings by using 2D visual stimuli presented on a blank screen under highly controlled experimental conditions. It has been argued that external validity is not warranted because in everyday life, learning and memory processes take place in interactive 3D contexts (e.g., [2]). Testing memory processes in real environments enhances external validity but reduces experimental control and experimental replicability. The advantage of VR is its high degree of experimental control, external validity and experimental replicability [1]. In addition, VR environments enable immersive and interactive experiences, which are not provided

**Competing interests:** The authors have declared that no competing interests exist.

by classical experimental settings. VR might bridge the gap between highly controlled laboratory settings and reality [2].

Since the seminal paper of Blascovich et al. [1], several studies have proven the benefits of using virtual reality in psychological research (e.g., [3–6]). To the best of our knowledge, no validated database of virtual objects has been developed yet. However, such stimuli are essential for studying processes such as e.g., learning and memory in VR. The goal of this article is to report the creation of 121 virtual objects that can directly be used in VR and the standardization of these stimuli in terms of name, familiarity, and visual complexity. The tested norms are in agreement with different approaches used to validate visual stimulus material (see, e.g., [7, 8]).

In sum, we aimed to create, standardize, and share a first set of virtual objects. Open science not only makes scientific work more visible and impactful but also increases efficiency, transparency, and transfer of knowledge. When stimulus sets are standardized and shared, they can be a powerful resource for other researchers to use in VR settings. For this reason, the present set of 3D objects is available online on the Open Science Framework (OSF). To our knowledge, this is the first open database of virtual objects. Such databases are extremely relevant to the growing trend of VR research in psychological sciences.

## Methods

### Participants

To standardize the virtual objects, data from an English-speaking sample was recruited from Amazon's Mechanical Turk. We preregistered the study on the OSF (https://osf.io/mc2ny). Ninety-six participants completed the online survey in exchange for $3. One participant took part in the survey two times. The duplicate was removed, and only the first completed survey was included in the final dataset. To restrict the sample to native English speakers, responses from non-US participants (12 participants) were removed. The remaining sample was comprised of 83 participants (51.8% female) with an average age of 37.5 years (SD = 12.7), 100% of whom were US residents. In terms of occupation, they were classified as sales (21.7%), technicians (18.1%), service sector (16.9%), academic profession (10.8%), executives (6%), students (4.8%), temporary staff (3.6%), social services (2.4%), or other (15.7%). The ethics committee of the Swiss Distance University Institute (2019-04-00002) approved the study, which was conducted according to the principles expressed in the Declaration of Helsinki.

### Materials

**Virtual objects.** Initially, ideas for virtual objects were gathered through discussions within the research group and aligned with pre-existing 2D databases. We selected inanimate objects that were common in daily usage and were not rotationally symmetric. Rotational *a*symmetric objects are objects that look differently in each distinct orientation (e.g., a cup with one handle but not a cup with two handles). This characteristic is for example needed in VR experiments testing orientation recall where subjects have to reinstate the original orientation of objects. A set of 121 objects belonging to categories such as furniture, electronic devices, or food, were built in Maya versions 18 and 19 (Autodesk). The textures of the objects were created in Substance Painter (Adobe). For each object, three to four texture maps (i.e., the BaseColor map, the normal map, the occlusion/roughness/metallic map, and the alpha map) were created and exported as PNG with a resolution of 2048 x 2048 pixels. Objects were exported from Maya to FBX and OBJ format and rendered online using the Unreal Engine 4 application (version 4.21.2, Epic Games). The objects (including textures) can be found on the OSF (https://osf.io/q658a/). The objects were designed to be changed in color in the Unreal

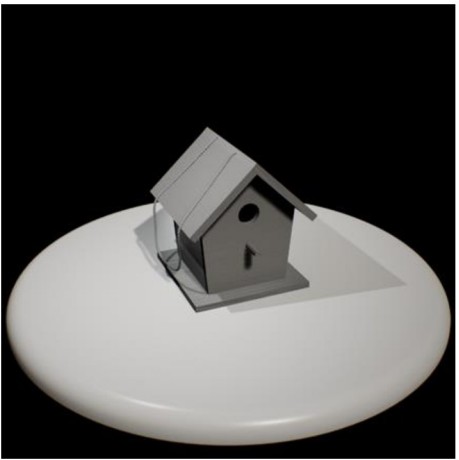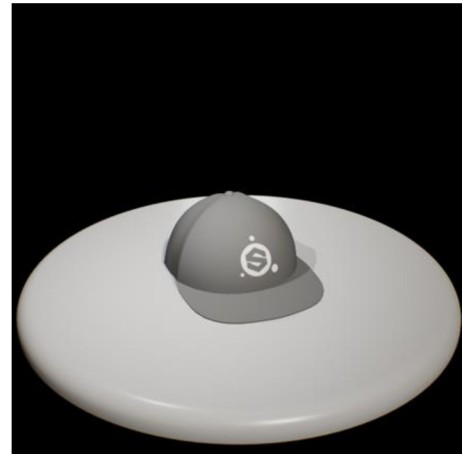

**Fig 1. Examples of stimuli as presented in the online survey.**

Engine. Some of the textures own an alpha texture to highlight details. For example, the cap's logo (see Fig 1) remains white when the alpha texture is applied. The objects and textures can be imported directly into the Unreal Engine, where the textures have to be connected into one material. The material will then be assigned to the corresponding virtual object. Examples of objects are presented in Fig 1.

For the standardization procedure, the virtual objects were converted to video format. After rendering the objects into AVI video files using Unreal Engine 4 application (pixel resolution: 1920 x 1080, frame rate: 60 fps, length: 4 seconds), they were converted into mp4 files for presentation in the online survey. Objects were presented in grayscale. The videos can also be found on the OSF (https://osf.io/q658a/).

*Name of virtual objects.* Following the procedure of Brodeur et al. [7], we asked participants to identify virtual objects with the following question: "Identify the object as briefly and unambiguously as possible by writing only one name, the first name that comes to mind. The name can be composed of more than one word."

*Familiarity.* Participants were asked to rate the level to which they are familiar with the object on a 5-point Likert scale ranging from *not familiar at all* to *very familiar*.

*Visual complexity.* Participants were asked to subjectively rate the level to which the object appears to be complex in terms of the quantity of details and the number of angles on a 5-point Likert scale ranging from *not complex at all* to *very complex*.

## Procedure

Participants completed the survey online using the freely available open-source software Lime-Survey (www.limesurvey.org). They were first asked to report informed consent, and then they reported age, gender, employment, country of origin, and country of residence. Moreover, they were certified to be at least 18 years of age and that they understand written English well. Virtual objects were then presented one at time with automatic playback of a four-second video showing all sides of the object in grayscale. It was possible to replay the video. Next, participants were asked to type the name of the object (name agreement task) and then provided their ratings of familiarity and of visual complexity. Participants could move to the next object at their own pace, and objects were presented in a randomized order across participants. The entire validation procedure lasted about 30 minutes.

## Analyses

Raw data as well as an Excel spreadsheet containing specific details related to each virtual object are available on the OSF (https://osf.io/q658a/). For each object, we report the modal name, name agreement, and mean ratings for familiarity and visual complexity.

## Modal name and name agreement

The name or combination of names given by the highest number of participants was considered the modal name of the object. Before determining modal names, individual entries were revised as follows:

1. Entries like "I don't know" or "no idea" were coded as DKO (don't know object).

2. Misspellings were corrected.

3. Abbreviated words were written out (e.g., "television" instead of "TV").

4. Conjunctions like "and," "or," and "with" were discarded.

5. Adjectives describing a state or a feature that is irrelevant to the identity of the object were also discarded (e.g., "cap" instead of "stylish cap"). Adjectives were not removed if they provided relevant information regarding the nature, shape, or function of the object (e.g., "wireless phone" or "artificial banana").

6. Composite names with a rearranged word order (e.g., "bowl of fruit" and "fruit bowl") were considered as the same name.

Once the modal name was identified, all entries that contained the modal name as previously defined (see above) were recoded with the modal name. For example, the responses "bottle opener" and "bottle cap opener" were recoded to the modal name "opener". This strategy was adopted to prevent single entries with basically the same meaning from gaining too much weight in the analyses and thus confound potentially high name agreement. The name agreement measure provides information about which objects elicit rather homogenous names and which objects are named rather inconsistently.

## Familiarity and visual complexity

The ratings for familiarity and visual complexity were computed by averaging scores on the 5-point scale and calculating standard deviations.

## Results

Table 1 summarizes the agreement and ratings for name, familiarity, and visual complexity, and Fig 2 shows their histograms.

**Table 1. Agreement and rating for name, familiarity and visual complexity.**

|  | Descriptive statistics | | | |
|---|---|---|---|---|
| **Variables** | **Mean** | **SD** | **Min** | **Max** |
| **Modal name agreement** | 74% | 22.8% | 20.5% | 100% |
| **DKO** | 0.55% | 1.43% | 0% | 7.20% |
| **Familiarity** | 4.37 | 0.42 | 2.77 | 4.80 |
| **Visual complexity** | 2.42 | 0.45 | 1.64 | 3.59 |

DKO = Don't know object. SD = Standard deviation.

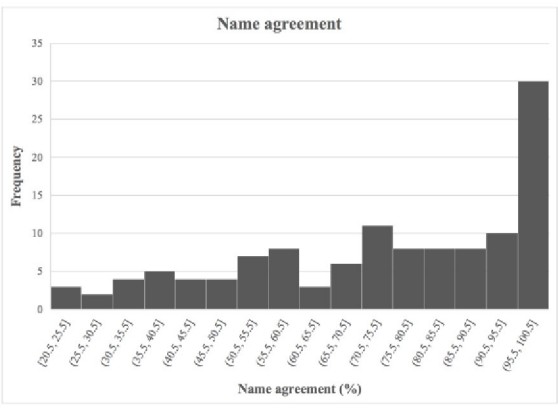
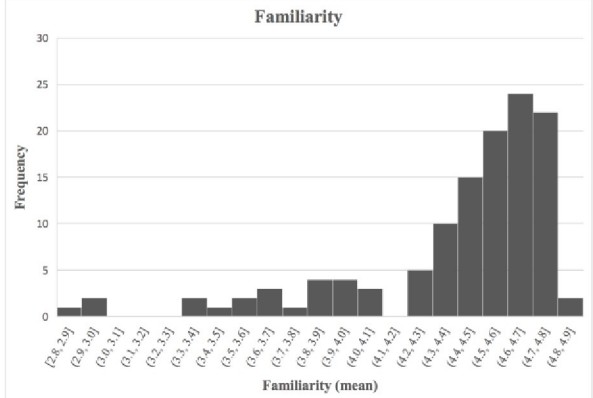
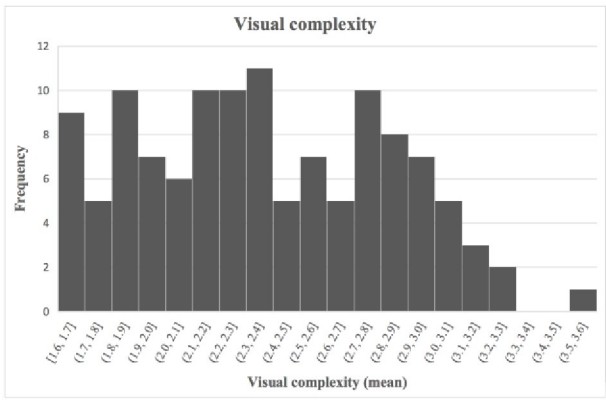

**Fig 2. Histograms of frequencies of norms.**

### Name of the virtual objects

The name of the virtual object was defined as the name given by the highest number of participants. Overall, 74% (SD = 22.8%) of participants agreed on the chosen names.

### Familiarity

Overall, the objects were rated as rather familiar ($M$ = 4.37, $SD$ = 0.42), as shown in Table 1.

### Visual complexity

Overall, the objects were rated as having medium complexity ($M$ = 2.42, $SD$ = 0.45), as shown in Table 1.

## Discussion

VR technology is a promising tool for bridging the gap between experimental control and external validity in psychological research. Sets of stimuli are valuable for development and validation of experimental material. Several sets of stimuli have been developed and validated, but virtual objects that can be directly used in a VR environment have not yet been made accessible. The goal of this study was to develop and validate the first normative database of virtual objects. The stimuli and stimulus-specific norms for each normative dimension are made available online on the OSF to facilitate access to experimental material for VR.

In this study, name agreement was 74% across all virtual objects and ranged from 20.50% to 100%. Thus, the percentage of name agreement was comparable to the percentage of name agreement found in studies using 2D line-drawn pictures (72%–85% depending on the language used [9]) and in studies using 2D photographs (64%; [10]). It is important to consider that recoding entries with more precise modal names to the modal names as we chose led to higher name agreement by eliminating potential alternative names. We argue that as long as the function of the object remains the same, it is acceptable to neglect additional details to prevent an artificial reduction in name agreement. However, by using this procedure, we might not have captured the diversity of the stimuli, which in turn represents the diversity of real-world objects.

Furthermore, the virtual objects in the present study were predominantly everyday objects, presented with a simple and unambiguous design, which would likely lead to high name agreement. The DKO rate of only 0.55% emphasizes that the objects were easily recognizable. In the second phase of the project conducted by Brodeur et al. [7], which added 930 new normative photos to the existing database, name agreement was substantially lower (mean of 58%) than in the first phase of the project. The authors argue that adding more stimuli generally leads to a reduction in name agreement because more uncommon objects have to be included. With only 121 objects, we have a relatively small database, which may further explain the high percentage of name agreement.

The average ratings of familiarity and visual complexity were 4.37 and 2.42 (both out of 5), respectively. It is not surprising that the mean familiarity rating was rather high since everyday objects were selected for this project. The score is comparable with the mean rating of familiarity (4.0 out of 5) found by Brodeur et al. [10], who also used common objects in their validation study. However, the average familiarity score reported by Snodgrass and Vanderwart [8] was numerically lower than the one we found. It is conceivable that 3D objects, as well as photos, correspond more with what we can see and touch in real life and thus lead to the impression of higher familiarity.

The average score obtained for visual complexity, which aligns with our expectations, might reflect the simple and colorless design of the virtual objects. Interestingly, it is consistent with the visual complexity score of 2.4 for photo stimuli reported by Brodeur et. al. [10]. The authors argue that although photos include more details than artificially designed objects, photo stimuli are more similar to what subjects perceive in everyday life. Indeed, the visual complexity score for line-drawn pictures reported in the database of Snodgrass and Vanderwart [8] was 3.0, which is numerically higher than that in the present study. To summarize, the set of virtual objects is comparable to pre-existing databases with 2D stimuli in terms of name agreement, familiarity rating, and visual complexity rating.

There are some limitations of the database and the norming procedure. A larger number of virtual objects should be developed and evaluated in future research. Moreover, future research would benefit from examining other dimensions of the database, such as object category or object agreement (i.e., the extent to which the object is similar to the one imagined by the subject). In the current validation study, objects were presented in grayscale and centered in the middle of a table. Color of objects can be changed, and objects can be presented in other (richer) contexts, however the relationship between color, context and ratings is unknown. It would be important to further investigate these aspects.

Although open source databases are growing in popularity, they have not yet achieved widespread use. In a recent review, McKiernan et al. [11] illustrated that open research is associated with increased citations, media attention, potential collaborators, job opportunities, and funding opportunities. A large majority of researchers support open science and contribute to efficient and transparent research practices. It is important to also make experimental

tools and stimulus material accessible to third parties to ensure a standardized procedure for research experiments.

Here, we present a database of 121 virtual objects. The objects were created specifically to help experimental psychologists who investigate perception and memory in virtual reality. Yet another possibility is to use these everyday objects to create virtual environments (e.g., in social or clinical psychology). The virtual objects have been normed for name, familiarity and visual complexity, thus allowing researchers to use this information as experimental variables or as control variables (to avoid any confounding effects). Finally, the present results indicate that the virtual objects are valid and may benefit future research in different fields of knowledge. We hope that the virtual objects will be useful for researchers adopting VR environments to test their research questions.

## Supporting information

**S1 Data. Original data.**
(XLSX)

**S2 Data. Ratings for individual objects.**
(XLSX)

## Author Contributions

**Conceptualization:** Corinna S. Martarelli.

**Data curation:** Simona G. Pacozzi.

**Formal analysis:** Simona G. Pacozzi.

**Investigation:** Corinna S. Martarelli.

**Methodology:** Corinna S. Martarelli.

**Project administration:** Corinna S. Martarelli.

**Resources:** Corinna S. Martarelli.

**Software:** Deian Popic.

**Supervision:** Corinna S. Martarelli.

**Validation:** Corinna S. Martarelli.

**Visualization:** Deian Popic, Simona G. Pacozzi.

**Writing – original draft:** Simona G. Pacozzi, Corinna S. Martarelli.

**Writing – review & editing:** Deian Popic, Simona G. Pacozzi, Corinna S. Martarelli.

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
