## [Decision Letter · Decision Letter 0]

15 Jun 2020

PONE-D-20-11816

Database of virtual objects to be used in psychological research

PLOS ONE

Dear Dr. Martarelli,

Thank you for submitting your manuscript to PLOS ONE. After careful consideration, we feel that it has merit but does not fully meet PLOS ONE’s publication criteria as it currently stands. Therefore, we invite you to submit a revised version of the manuscript that addresses the points raised during the review process.

We look forward to receiving your revised manuscript.

Kind regards,

Haoran Xie

Academic Editor

PLOS ONE

Journal Requirements:

Reviewers' comments:

Reviewer's Responses to Questions

**Comments to the Author**

1. Is the manuscript technically sound, and do the data support the conclusions?

Reviewer #1: Yes

Reviewer #2: Yes

2. Has the statistical analysis been performed appropriately and rigorously? 

Reviewer #1: Yes

Reviewer #2: Yes

3. Have the authors made all data underlying the findings in their manuscript fully available?

Reviewer #1: Yes

Reviewer #2: Yes

4. Is the manuscript presented in an intelligible fashion and written in standard English?

Reviewer #1: Yes

Reviewer #2: Yes

5. Review Comments to the Author

Reviewer #1: Database of virtual objects to be used in psychological research

This study features a downloadable dataset of 3D object sets for virtual realty research, an important fast-growing research trend. As authors stated, the coming of this dataset is timely, and its availability for free download also speeds up the dissemination of knowledge and contributing to its widespread usage. In addition, several indices (N=80), including object familiarities, complexity, and name agreement, were provided for cross-database checks.

Overall, I highly agree with authors’ point on making these AR/VR-compatible stimuli freely accessible, with raters’ norms data, provided a good start point toward wider usage and further improvement, stimulation, and growing progress. That being said, I do have several suggested edits and 1~2 wishes that would require authors’ responses.

Line 47 (pp. 3): control, external validity, and (misplace comma) experimental replicability [1]

Line 64-65 (pp. 4): Such databases are extremely relevant, as VR technology is becoming more popular in psychological research. Relevant to what? “to the growing popularity/trend of VR research in psychological sciences”.

Line 69: recruited from Amazon’s website Mechanical Turk.

Line 78: The local ethics 78 committee approved the study, which … Could the name of the IRB and document number being stated here, if possible?

Line 84: were not "rotationally symmetric”….not sure what it meant here for the name of the VR stimulus set. After viewing the stimulus mp4, I guess it meant to represent the rotation of stimuli on the table, so could be better comprehended to add reference to the mp4.

Line 89: when I first saw the “FBX” format, I wondered if FBX is a standard version for VR stimuli, and googled online (and found this one: https://experience.briovr.com/blog/obj-and-fbx-files-for-virtual-reality-augmented-reality/). I guess the authors would also like to add some references of “FBX” for naive readers.

Line 197: "The average ratings of familiarity and visual complexity were 4.37 and 2.42”. I would suggest the addition of (both out of 5) immediately after the numbers, and make sure that the scale were consistent across mentioned studies in this paragraph (e.g., Brodeur [7] and Snodgrass and Vanderwart [8].The reason I raised this concern is that as the authors mentioned, in line 213-214, that "texture can appear artificial in drawings, leading to ambiguity and creating the impression of higher visual complexity.” seem counterintuitive at first glance. Because the Snodgrass et al. stimuli were mostly black-line drawings, and not clear whether the so-called “textures” was present in affecting the raters around 80'.

Suggestion 1: after downloading the video, I found the choice of colorless object and the smoothness of the object (e.g., Bananas, wither 1 or many, were not that smooth in rendered video). I guess as the 1st release, there are definitely rooms for improvement for the future release. I only hope that it will not be 24 or 32 years apart, such as the Snowgrass et al., and the subsequent color version (Rossion et al. 2004) https://journals.sagepub.com/doi/abs/10.1068/p5117 and (Moreno et al. 2012) https://journals.plos.org/plosone/article?id=10.1371/journal.pone.0037527#

Suggestion 2: To concur with the authors on the benefits of open science, I like to suggest one further step: a road map on how to best use this uploaded VR stimuli. E.g., to create a psychological VR experiment. First, I guess a VR headset would be required, and then code with certain, possibly open-source software, and then adopting material, and investigate the effect? How could the research make the best use of such stimuli? Some road map suggestion would be highly desired, or is there already some publication that could be of reference?

Reviewer #2: The purpose of this research is clear, the design and procedure are straight forward, and the methodology is consistent with some other researches in this area. These are the strength of this paper. In terms of significance, virtual reality does play a more and more important role in psychological experiments nowadays. Therefore, building good databases, especially the first one with 3-D objects, is truly important work.

Here are the questions need some clarification. First. the “database” is small with only 121 objects. Especially many of them have more than one version.

About the participants. 12 non-US participants were removed because they were not “the target population”. Are the participants removed by their citizenship or English proficiency?

Finally, this paper aimed to “create, standardize, and share the first set of objects” for psychological experiments. The standardization might need some more elaboration.

In sum, the issue is important and the methodology is fine. However, the database is small, as well as there are some conceptual concerns to be clarified.

6. PLOS authors have the option to publish the peer review history of their article (what does this mean?). If published, this will include your full peer review and any attached files.

Reviewer #1: Yes: Chun-Chia Kung

Reviewer #2: No

---

## [Author Response · Author response to Decision Letter 0]

22 Jun 2020

Dear Prof. Xie, dear Editor

We thank you very much for sending us the reviews for our manuscript entitled “Database of virtual objects to be used in psychological research”. We thank you for your comments and we are happy that you think that the study has the potential to make enough of a contribution on its own. In agreement with the reviewers’ suggestions, we have developed more deeply some methodological aspects and we have extended the limitation section.

We greatly appreciate the opportunity to submit this revision of our manuscript and we have revised the manuscript in accordance with the reviewers’ recommendations, which we found to be clear, helpful, and very constructive. 

We hereby submit the revised version of the manuscript and accompanying reply letter explaining how we addressed the reviewers’ comments. 

We hope that we have satisfactorily addressed their suggestions and thank you for considering this paper for publication in PLoS ONE. Please do not hesitate to contact us if you have any questions.

Yours sincerely and on behalf of my coauthors,

Corinna Martarelli

 

Resubmission of the manuscript entitled “Database of virtual objects to be used in psychological research” (PONE-D-20-11816)

Point-by-point reply

Note: the revised passages are written in blue in the manuscript. 

Reviewer 1:

This study features a downloadable dataset of 3D object sets for virtual reality research, an important fast-growing research trend. As authors stated, the coming of this dataset is timely, and its availability for free download also speeds up the dissemination of knowledge and contributing to its widespread usage. In addition, several indices (N=80), including object familiarities, complexity, and name agreement, were provided for cross-database checks.

Overall, I highly agree with authors’ point on making these AR/VR-compatible stimuli freely accessible, with raters’ norms data, provided a good start point toward wider usage and further improvement, stimulation, and growing progress. That being said, I do have several suggested edits and 1~2 wishes that would require authors’ responses.

Reply: We thank Prof. Kung very much for his helpful and very constructive comments. Further we greatly appreciate the general positive assessment. We tried to address each of the points raised, and suggestions made have been very helpful in further developing our paper. Please find below how we addressed your points. 

Reviewer 1:

Line 47 (pp. 3): control, external validity, and (misplace comma) experimental replicability [1]

Reply: Thank you, we removed the comma.

Reviewer 1:

Line 64-65 (pp. 4): Such databases are extremely relevant, as VR technology is becoming more popular in psychological research. Relevant to what? “to the growing popularity/trend of VR research in psychological sciences”.

Reply: In line with your suggestion we rewrote the sentence that now reads as follows:

Such databases are extremely relevant to the growing trend of VR research in psychological sciences. 

Reviewer 1:

Line 69: recruited from Amazon’s website Mechanical Turk.

Reply: We removed website and now write “recruited from Amazon’s Mechanical Turk”.

Reviewer 1:

Line 78: The local ethics 78 committee approved the study, which … Could the name of the IRB and document number being stated here, if possible?

Reply: Thank you for drawing our attention to this issue, we now implemented the missing information. The sentence now reads as follows:

The ethics committee of the Swiss Distance University Institute (2019-04-00002) approved the study, which was conducted according to the principles expressed in the Declaration of Helsinki.

Reviewer 1:

Line 84: were not "rotationally symmetric”….not sure what it meant here for the name of the VR stimulus set. After viewing the stimulus mp4, I guess it meant to represent the rotation of stimuli on the table, so could be better comprehended to add reference to the mp4.

Reply: We thank Reviewer 1 for directing our attention to this issue. Rotational symmetry is the property of an object when it looks the same after some degree of rotation. We wanted to avoid this rotational symmetry, thus we created for example a cup with one handle and not a cup with two handles. Rotational asymmetry is important when for example testing object orientation recall in a virtual environment. In the revised version of the manuscript we now write:

Rotational asymmetric objects are objects that look differently in each distinct orientation (e.g., a cup with one handle but not a cup with two handles). This characteristic is for example needed in VR experiments testing orientation recall where subjects have to reinstate the original orientation of objects.

Reviewer 1:

Line 89: when I first saw the “FBX” format, I wondered if FBX is a standard version for VR stimuli, and googled online (and found this one: https://experience.briovr.com/blog/obj-and-fbx-files-for-virtual-reality-augmented-reality/). I guess the authors would also like to add some references of “FBX” for naive readers.

Reply: Thank you for this comment! The most up to date file format for VR is FBX, however OBJ files are also universal. These are the two universal file formats for virtual objects (as for example PNG and JPG for 2D stimuli). We acknowledge the concern of the Reviewer and decided to additionally export the files also in OBJ format. We implemented this information in the revised version of the manuscript and uploaded the OBJ export to the database on the OSF. Thank you, we think that this is a fine addition to our database that extends its accessibility. 

Reviewer 1:

Line 197: "The average ratings of familiarity and visual complexity were 4.37 and 2.42”. I would suggest the addition of (both out of 5) immediately after the numbers, and make sure that the scale were consistent across mentioned studies in this paragraph (e.g., Brodeur [7] and Snodgrass and Vanderwart [8].The reason I raised this concern is that as the authors mentioned, in line 213-214, that "texture can appear artificial in drawings, leading to ambiguity and creating the impression of higher visual complexity.” seem counterintuitive at first glance. Because the Snodgrass et al. stimuli were mostly black-line drawings, and not clear whether the so-called “textures” was present in affecting the raters around 80'.

Reply: We acknowledge your suggestion and added “out of 5” in the revised version of the manuscript. Yes, the studies mentioned in the paragraph also used 5-point Likert scales. The sentence was indeed ambiguous. We removed the sentence. Thank you! 

Reviewer 1:

Suggestion 1: after downloading the video, I found the choice of colorless object and the smoothness of the object (e.g., Bananas, wither 1 or many, were not that smooth in rendered video). I guess as the 1st release, there are definitely rooms for improvement for the future release. I only hope that it will not be 24 or 32 years apart, such as the Snowgrass et al., and the subsequent color version (Rossion et al. 2004) https://journals.sagepub.com/doi/abs/10.1068/p5117 and (Moreno et al. 2012) https://journals.plos.org/plosone/article?id=10.1371/journal.pone.0037527#

Reply: Thank you for this comment. In the revised version of the manuscript we now make more explicit that the object’s color can be changed in the Unreal Engine. We now write:

The objects were designed to be changed in color in the Unreal Engine. Some of the textures own an alpha texture to highlight details. For example, the cap’s logo (see Figure 1) remains white when the alpha texture is applied. The objects and textures can be imported directly into the Unreal Engine, where the textures have to be connected into one material. The material will then be assigned to the corresponding virtual object.

We decided to keep the color constant (grayscale) in the validation procedure. Yes, hopefully we do not have to wait 24-32 years for a next VR database and / or validation study. We now added the following sentence in the limitation section:

In the current validation study, objects were presented in grayscale and centered in the middle of a table. Color of objects can be changed, and objects can be presented in other (richer) contexts, however the relationship between color, context and ratings is unknown. It would be important to further investigate these aspects.

Reviewer 1:

Suggestion 2: To concur with the authors on the benefits of open science, I like to suggest one further step: a road map on how to best use this uploaded VR stimuli. E.g., to create a psychological VR experiment. First, I guess a VR headset would be required, and then code with certain, possibly open-source software, and then adopting material, and investigate the effect? How could the research make the best use of such stimuli? Some road map suggestion would be highly desired, or is there already some publication that could be of reference?

Reply: Many thanks for this suggestion. We do think that there are many ways as these virtual objects could be used. In our lab we are for example using the virtual objects to investigate memory recall with continuous measures (recall of location, color and orientation of the objects). The VR environment allows for a high degree of experimental control, i.e., the accurate measurement of location, orientation, and color of virtual objects while allowing participants to engage with their environment in a realistic manner. However, other research groups might need objects to fill in virtual rooms. Then they could use these objects, without the objects being the central part of their research question. In the revised version of the manuscript we now write:

The objects were created specifically to help experimental psychologists who investigate perception and memory in virtual reality. Yet another possibility is to use these everyday objects to create virtual environments (e.g., in social or clinical psychology). The virtual objects have been normed for name, familiarity and visual complexity, thus allowing researchers to use this information as experimental variables or as control variables (to avoid any confounding effects). Finally, the present results indicate that the virtual objects are valid and may benefit future research in different fields of knowledge. We hope that the virtual objects will be useful for researchers adopting VR environments to test their research questions.

Reviewer #2: 

The purpose of this research is clear, the design and procedure are straight forward, and the methodology is consistent with some other researches in this area. These are the strength of this paper. In terms of significance, virtual reality does play a more and more important role in psychological experiments nowadays. Therefore, building good databases, especially the first one with 3-D objects, is truly important work.

Reply: We are happy that Reviewer 2 thinks the present work is a valuable contribution. Please find below how we addressed the comments of Reviewer 2.

Reviewer #2: 

Here are the questions need some clarification. First. the “database” is small with only 121 objects. Especially many of them have more than one version.

Reply: We agree that the database is small. However, we think that it is relevant to the growing VR research to make this first database freely accessible. We acknowledge the concern of Reviewer 2 and extend the limitation section that now reads as follows: 

There are some limitations of the database and the norming procedure. A larger number of virtual objects should be developed and evaluated in future research. Moreover, future research would benefit from examining other dimensions of the database, such as object category or object agreement (i.e., the extent to which the object is similar to the one imagined by the subject). In the current validation study, objects were presented in grayscale and centered in the middle of a table. Color of objects can be changed, and objects can be presented in other (richer) contexts, however the relationship between color, context and ratings is unknown. It would be important to further investigate these aspects. 

Reviewer #2: 

About the participants. 12 non-US participants were removed because they were not “the target population”. Are the participants removed by their citizenship or English proficiency?

Reply: Participants were removed by their citizenship (all participants confirmed that they understand written English well). We wanted to restrict the study to native English speakers. Studies comparing US and Indian MTurk participants show that in language-based tasks US MTurk participants give higher quality responses than Indian MTurk participants (e.g., Kazai et al., 2012). It has been suggested that non-native English speakers should be avoided when the task is language based (e.g., Hauser et al., 2018). Following this recommendation, we preregistered our study and stated that we will exclude responses from non-US participants. Please see our preregistration here: https://osf.io/mc2ny

Kazai, G., Kamps, J., & Milic-Frayling, N. (2012). The face of quality in crowdsourcing relevance labels: Demographics, personality and labeling accuracy. In Proceedings of the 21st ACM international conference on Information and knowledge management.

Hauser, D. J., Paolacci, G., Chandler, J. (2019). Common concerns with Mturk as a participant pool: Evidence and solutions. In Handbook of Research Methods in Consumer Psychology, edited by Kardes, F. R., Herr, P. M., & Schwarz, N. New York: Routledge.

Following the suggestion of Reviewer 2, we now make clear that we removed participants by their citizenship. In the revised version we now write:

To restrict the sample to native English speakers, responses from non-US participants (12 participants) were removed.

Reviewer #2: 

Finally, this paper aimed to “create, standardize, and share the first set of objects” for psychological experiments. The standardization might need some more elaboration.

In sum, the issue is important and the methodology is fine. However, the database is small, as well as there are some conceptual concerns to be clarified.

Reply: Thank you for directing our attention to this issue. We agree that the database is small and that the standardization might need some more elaboration. We addressed these issues in the limitation section of the manuscript, please see our answer to your comment above.

---

## [Decision Letter · Decision Letter 1]

10 Aug 2020

Database of virtual objects to be used in psychological research

PONE-D-20-11816R1

Dear Dr. Martarelli,

We’re pleased to inform you that your manuscript has been judged scientifically suitable for publication and will be formally accepted for publication once it meets all outstanding technical requirements.

Kind regards,

Haoran Xie

Academic Editor

PLOS ONE

Additional Editor Comments (optional):

Reviewers' comments:

Reviewer's Responses to Questions

**Comments to the Author**

1. If the authors have adequately addressed your comments raised in a previous round of review and you feel that this manuscript is now acceptable for publication, you may indicate that here to bypass the “Comments to the Author” section, enter your conflict of interest statement in the “Confidential to Editor” section, and submit your "Accept" recommendation.

Reviewer #1: (No Response)

Reviewer #2: All comments have been addressed

2. Is the manuscript technically sound, and do the data support the conclusions?

Reviewer #1: Yes

Reviewer #2: Yes

3. Has the statistical analysis been performed appropriately and rigorously? 

Reviewer #1: Yes

Reviewer #2: Yes

4. Have the authors made all data underlying the findings in their manuscript fully available?

Reviewer #1: Yes

Reviewer #2: Yes

5. Is the manuscript presented in an intelligible fashion and written in standard English?

Reviewer #1: Yes

Reviewer #2: Yes

6. Review Comments to the Author

Reviewer #1: Most of my (as of Reviewer 1) comments were addressed, and suggested things added (such as the new virtual objects OBJ file format, other than the original only FBX format, among other points). Overall, I am happy for their improvementss and correspondences, and therefore have no further Q/suggestions. Also sorry for the belated response.

Reviewer #2: (No Response)

7. PLOS authors have the option to publish the peer review history of their article (what does this mean?). If published, this will include your full peer review and any attached files.

Reviewer #1: **Yes: **Chun-Chia Kung

Reviewer #2: No

---

## [Editor Report · Acceptance letter]

26 Aug 2020

PONE-D-20-11816R1 

Database of virtual objects to be used in psychological research 

Dear Dr. Martarelli:

I'm pleased to inform you that your manuscript has been deemed suitable for publication in PLOS ONE. Congratulations! Your manuscript is now with our production department. 

Kind regards, 

on behalf of

Professor Haoran Xie 

Academic Editor

PLOS ONE